# Mast Cells in the Auditory Periphery of Rodents

**DOI:** 10.3390/brainsci10100697

**Published:** 2020-10-01

**Authors:** Agnieszka J. Szczepek, Tatyana Dudnik, Betül Karayay, Valentina Sergeeva, Heidi Olze, Alina Smorodchenko

**Affiliations:** 1Department of Otorhinolaryngology, Head and Neck Surgery, Charité-Universitätsmedizin Berlin, Corporate Member of Freie Universität Berlin, Humboldt-Universität zu Berlin, and Berlin Institute of Health, 10117 Berlin, Germany; tatyana.dudnik@charite.de (T.D.); betuel.karayay@charite.de (B.K.); heidi.olze@charite.de (H.O.); 2Department of Medical Biology with Course of Microbiology and Virology, Chuvash State University, 428034 Cheboksary, Russia; kafedra-biology@yandex.ru; 3Department of Human Medicine, MSH Medical School Hamburg, University of Applied Sciences and Medical University, 20457 Hamburg, Germany; alina.smorodchenko@medicalschool-hamburg.de

**Keywords:** auditory periphery, mast cell, cochlea

## Abstract

Mast cells (MCs) are densely granulated cells of myeloid origin and are a part of immune and neuroimmune systems. MCs have been detected in the endolymphatic sac of the inner ear and are suggested to regulate allergic hydrops. However, their existence in the cochlea has never been documented. In this work, we show that MCs are present in the cochleae of C57BL/6 mice and Wistar rats, where they localize in the modiolus, spiral ligament, and stria vascularis. The identity of MCs was confirmed in cochlear cryosections and flat preparations using avidin and antibodies against c-Kit/CD117, chymase, tryptase, and FcεRIα. The number of MCs decreased significantly during postnatal development, resulting in only a few MCs present in the flat preparation of the cochlea of a rat. In addition, exposure to 40 µM cisplatin for 24 h led to a significant reduction in cochlear MCs. The presence of MCs in the cochlea may shed new light on postnatal maturation of the auditory periphery and possible involvement in the ototoxicity of cisplatin. Presented data extend the current knowledge about the physiology and pathology of the auditory periphery. Future functional studies should expand and translate this new basic knowledge to clinics.

## 1. Introduction

Mast cells are part of the innate immune system of myeloid lineage and are derived from the bone marrow, from where they home to the mucosal tissues, or from the fetal liver, from where they home to the connective tissues [1,2]. Mature MCs are present in peripheral organs such as the lung [3], intestine [4], heart [5], eye [6], brain [7], and in many other structures [8]. MCs produce a variety of unique proteins and substances used for their identification [9]. Some of them, such as c-Kit receptor for MGF (Mast Cell Growth Factor, also called Stem Cell Factor), designated CD117, are expressed on the cell surface. Some others are localized in the mast cell secretory granules. The granule-localized substances include heparin, mast cell tryptases, mast cell chymases, and other proteases [10,11], immune mediators, and neuromodulators [11,12,13,14].

In addition to their well-explored role in allergic responses [15], MCs are involved in the regulation of vascular permeability [16], autoimmune diseases [17], and various stress-related and neurodegenerative conditions, including depression [18] and Alzheimer’s disease [19]. This involvement is possible due to the capacity of MCs to interact with an array of cell types, including, but not limited to, lymphocytes [20], neurons [21], microglia [22], and epithelium [23]. The diversity of stimuli to which MCs are capable of reacting to, and the wealth of cell types that can respond to the MC mediators, make them potent regulators of local and systemic physiological and pathological reactions.

In the sensory organs, the presence of MCs has been reported in the nasal mucosa [24,25], in the ocular tissues [26], and the gustatory tissues [27]. In the inner ear, MCs were detected in the endolymphatic sac of humans and guinea pigs [28,29]. In addition, degranulated MCs were observed in the lumen of the endolymphatic sac of guinea pigs after allergen-mediated induction of hypersensitivity type I [30] that was functionally associated with endolymphatic hydrops.

To our best knowledge, no published research has focused on the presence of MCs in the cochlea up to now. We reasoned that this topic could be essential to explore to advance our understanding of the inner ear biology and pathology. Therefore, we initiated this study, in which we performed a systematic search for MCs.

## 2. Materials and Methods

### 2.1. The Animals

This study was carried out in accordance with the recommendations of the EU Directive 2010/63/EU on the protection of animals used for scientific purposes. The experimental protocol was approved by the Governmental Ethics Commission for Animal Welfare (LaGeso Berlin, Germany; approval numbers: T 0234/00, T 0292/16, and T 0235/18). Newborn Wistar rats (outbred animal strain) and C57BL/6 mice were purchased from the local animal facility (the research service facility within the Charité—Universitätsmedizin Berlin responsible for experimental animal breeding and keeping, animal transport, animal welfare, and veterinary services). The animals were of both genders and were 1, 3, 5, 7, 9 days old (total *n* = 120 for Wistar rats and *n* = 80 for C57BL/6 mice). The experiments with 30-day-old animals (each *n* = 20 for Wistar rats and C57BL/6 mice) were conducted in accordance with international guidelines and the 3R program (Reduce, Refine, Replace).

### 2.2. Cryosections

For the cryosections, the base of the skull with both temporal bones was prepared. The specimens were fixed overnight in 4% formalin at +4 °C, and the temporal bones from 30-day-old animals were additionally incubated in 20% EDTA solution pH 7.4 (Carl Roth GmbH + Co. KG, Karlsruhe, Germany) for three days at +4 °C, and washed in 0.1 M PBS. Subsequently, the tissue was incubated in 15% and 30% sucrose solution at +4 °C for three days. Next, the specimens were placed in an aluminum box containing the Optimal Cutting Temperature (OCT) tissue freezing medium (cat. #0201 08926, Leica, Wetzlar, Germany). The tissue was shock-frozen in liquid nitrogen and then kept at −20 °C. Then, 7–10 μM cryosections were prepared using cryostat Leica CM 3050S (Germany). The sections were stored at −20 °C until further processing.

### 2.3. Explant Cultures and Flat Preparations of the Cochleae

The explants were prepared as previously described [31]. Briefly, following decapitation, the cochlea was dissected from the temporal bone and placed under the stereoscope SteREO Discovery.V8 (Carl Zeiss, Germany) to isolate the membranous cochleae. After removing cartilage, bone capsule, stria vascularis, and spiral ligament, the membranous cochleae were divided into three equal parts: an apical, medial and basal. Each section contained modiolus, spiral limbus with spiral ganglion neurons, and the organ of Corti. Cochlear tissues were explanted in the 4-well culture dishes containing 500 µL of tissue culture medium (DMEM/F12; (cat. #21331-020, Gibco^®^, Karlsruhe, Germany), supplemented with 10% fetal bovine serum (FBS, cat. #S0113, Biochrom AG, Berlin, Germany), 2.5 M glucose (cat. #G8769, Sigma, Aldrich, Germany), insulin-transferrin–Na-selenite (cat. #11207500, Roche, Basel, Switzerland), penicillin G (cat. #A321-42, Biochrom AG, Berlin, Germany), and IGF-1 (#4326-RG R&D Systems, Wiesbaden, Germany). The culture was conducted in a humidified incubator at +37 °C and 5% CO_2_ for 24 h. The explants were fixed in 10% formalin (cat. #HT5011, Sigma-Aldrich, Darmstadt, Germany) for 40 min at room temperature (RT) and kept at +4 °C for future immunohistochemical/immunofluorescence (IHC/IF) staining. The explanted membranous cochlear tissues (flat preparations) were used for staining and scoring of MCs because of the complete view of the basilar membrane with the organ of Corti, spiral limbus, and the spiral ganglion neurons as opposed to the cryosections or paraffin sections that were prepared to achieve a cross-sectional view through the cochlea.

### 2.4. Exposure to Cisplatin

The cisplatin (*cis*-diamminedichloridoplatinum II, cat. #CAS 15663-27-1, Calbiochem, Merck, Darmstadt, Germany) was dissolved in dimethyl sulfoxide (DMSO) to achieve a concentration of 100 mg/mL. A stock solution (3.33 mM) was prepared by diluting the cisplatin at 1 mg/mL of the tissue culture media. Cochlear explants dissected from p5 (5th postnatal day) Wistar rats were cultured as described above for 24 h in the 4-well plates with cisplatin in concentrations of 10, 20, and 40 μM. Explants cultured without cisplatin served as negative controls.

### 2.5. Paraffin Embedding

For paraffin embedding, the isolated bases of skulls containing both temporal bones were fixed overnight in 4% formalin at +4 °C. The temporal bones from 30-day-old animals were additionally and subsequently incubated in 20% EDTA solution pH 7.4 (Carl Roth GmbH + Co. KG, Karlsruhe, Germany) for three days at +4 °C, washed in 0.1 M PBS, and embedded in paraffin using KD-500 tissue processor (Roundfin, China). Then, 10-μM-thick paraffin sections were prepared using Microm HM 355S (Leica), followed by deparaffinization in xylene and a series of ethanol before staining.

### 2.6. Immunofluorescent Staining

The specimens (cryo- and paraffin-embedded sections) were washed with 0.1 M PBS, permeabilized with 0.25% Triton X-100 (Fluka) in PBS and blocked with 4% normal goat serum (NGS, Jackson) to prevent the non-specific antibody binding for 60 min at RT, as previously described [32]. After this step, the specimens were incubated with the antibodies against the target molecule (for detailed information about the antibodies, see Table 1) for 30 min at +37 °C in a staining buffer containing 0.1 PBS, 0.1% Triton X-100, and 1% NGS. Next, the secondary antibodies conjugated with a fluorescent dye were applied for 60 min at RT. For the detection of heparin-containing granules, the specimens were incubated with avidin conjugated to AlexaFluor-488 (Invitrogen), whereas the nuclei were labeled with DAPI (Dianova, Hamburg, Germany). ProLong Gold Antifade Mountant was added to fix the coverslip in place and to prevent fluorescence fading. As a positive control, thymus tissues were used (data not shown). In negative controls, the primary antibody was omitted.

### 2.7. Light, Epifluorescent and Confocal Microscopy

Digital imaging was performed using an epifluorescence microscope EVOS FL Cell Imaging System (Thermo Fisher Scientific, Waltham, MA, USA) with ×10, ×20, and ×40 objectives. Light microscopy was performed on BX51 Olympus microscope (Tokio, Japan). Confocal images were made using confocal microscope Leica TCS SP5 (Leica Microsystems, Wetzlar, Germany) with immersion oil objectives ×40 and ×63. The Alexa-488, Alexa-594, and DAPI were excited using argon-laser (488 nm), helium-neon laser (543 nm), and blue diode laser (405 nm), respectively. The images were pseudocolored with RGB tools and reconstructed with ImageJ software (http://rsb.info.nih.gov/ij/). The quantitative image analyses (cell size) were performed using ImageJ software (https://imagej.nih.gov/).

### 2.8. Western Blot

The cochlear lysates were prepared with membranous cochlea by placing them in microcentrifuge tubes containing 80 µL RIPA buffer (Cell Signaling, #9806). The lysates were then clarified by centrifugation at 14.000 rpm for 10 min at +4 °C. The concentration of protein was determined using the Micro BCA protein assay kit (Thermo Fisher Scientific, Darmstadt, Germany, #23235). The aliquots containing 6 µg of total protein (p1, p3, p5, p7, p9) were mixed with Roti-Load sample loading solution (ROTH, #K929.1) and heated at 90 °C for 5 min in a Thermomixer comfort (Eppendorf Vertrieb, Hamburg, Germany), then transferred onto Novex WedgeWell 4–20% Tris-Glycine Mini Gels 12-well (Thermo Fischer Scientific, Darmstadt, Germany, #XP04202BOX) and 15-well Novex WedgeWell 4–20% Tris-Glycine Mini Gels (Thermo Fischer Scientific, Darmstadt, Germany #XP04205BOX) using a mini-SDS-PAGE system XCell SureLock Electrophoresis Cell (Invitrogen, Carlsbad, CA, USA #1287724-0959) at 130 V for 1 h and 40 min. The protein marker used was a PageRuler Plus Prestained Protein Ladder (Thermo Fischer Scientific, Darmstadt, Germany, #26619). After the electrophoresis, resolved proteins were transferred onto 0,45 µm Immobilon-P Transfer Membrane (Millipore, Darmstadt, Germany, #IPFL 000 10) using XCell II Blot Module (Invitrogen, #E19051) at 300 mA for 44 min (Biometra GmbH, Göttingen, Germany). The membranes were blocked with 5% skimmed milk powder solution prepared in PBS and containing 0.1% Tween 20 (Sigma-Aldrich, Taufkirchen, Germany, #524653-1EA) for one hour at RT followed by incubation for 2 h at RT with one of the primary rabbit antibodies anti-CD117 (Table 1). Following several washes in PBS 0.05% Tween 20 for 10 min, the secondary antibodies were added. The signal was detected by incubation of the blot with SuperSignal West Femto Maximum Sensitivity Substrate (ThermoScientific, #34095) and direct measurement of chemiluminescence by C-Digit scanner (LI-COR Biotechnology-GmbH, Bad Homburg vor der Höhe, Germany). The quantification was done with GelScan Pro V.6.0 software.

### 2.9. Statistical Analyses

Statistical analyses were performed using SigmaPlot software version 13.0 (Systat Software GmbH, Erkrath, Germany). The descriptive statistics were generated for the entire data set. The significance of differences between the means of cochlear MCs scored on different postnatal days was tested using Student’s t-test or one-way ANOVA; the alpha value was set to 0.05.

## 3. Results

### 3.1. Avidin-Positive Cells Are Present in the Cochleae of Mice and Rats

MCs were visualized in the cochlea using staining of cochlear paraffin sections derived from mice and rats with avidin-AlexaFluor-488 [33]. The avidin-positive, granulated cells were identified in modiolus and spiral limbus of both species (Figure 1). Occasionally, avidin-positive cells were visible in close proximity to the Reissner’s membrane (Figure 1C—mouse, arrow, and Figure 1E, rat, arrow) from the side of the scala vestibuli. No MCs were detected in or close to the organ of Corti (OC). No free scattered MCs (outside the tissue section) were seen; all MCs were located within the tissues. The bright-field microscopy (Figure 1F) and immunofluorescence (Figure 1G) show the presence of mast cells within the cochlear bone cavity.

### 3.2. The Avidin-Positive Cochlear Cells Express CD117

Next, the presence of transmembrane tyrosine-kinase receptor CD117 (c-kit) was investigated in avidin-positive cochlear cells of Wistar rats. Flat preparations of cochlear tissues were incubated with avidin conjugated with Alexa Fluor^®^ 488 and with rabbit IgG directed against CD117 followed by incubation with the secondary antibody against rabbit IgG conjugated with Alexa Fluor^®^ 594. The representative micrograph of the modiolus region verified the localization of CD117 signal on the avidin-positive cells (Figure 2A,B). The expression of CD117 antigen was consistent with the surface expression.

### 3.3. Cochlear MCs Express Tryptase, Chymase, and the High-Affinity Immunoglobulin Epsilon Receptor Subunit Alpha

The MCs produce tryptase and accumulate it in the granules. To determine if tryptase is present in the avidin-positive cells, tryptase-specific staining was performed with cochlear specimens obtained from p3 Wistar rats. The experiments confirmed the presence of tryptase within the cytoplasmic granule of avidin-positive cochlear cells, further confirming their identity as MCs (Figure 2C). An additional MC marker is chymase. Because of the lack of suitable reagents to perform the immunostaining in rat tissues, mouse cochlear tissues (flat preparations) were stained with antibodies against chymase. The staining confirmed that the avidin-positive cochlear MCs contain chymase. The chymase-positive cells were detected in the cochleae of p2 and p30 mice (Figure 2D). Another MC marker, high-affinity immunoglobulin epsilon receptor subunit alpha (FcεRIα), was also seen in the mouse cochlear flat preparations (Figure 3).

### 3.4. Levels of c-Kit/CD117 Protein in the Cochlea Change During the First Nine Postnatal Days

The level of 145 kDa CD117 protein in the membranous cochlear explants (containing basilar membrane, the organ of Corti, spiral limbus, and spiral ganglion neurons) was determined using Western blotting. The results demonstrated a statistically significant increase of CD117 in the cochleae of 5- and 7-day-old rats compared to the first, third, and the ninth day of life (Figure 4). As a positive control, the thymus from the same experimental animals was used. In contrast, CD117 was expressed in the thymus (control tissue containing MCs) on even levels during all studied post-developmental days.

### 3.5. The Number of MCs in Cochlear Explants Decreases During Postnatal Maturation of Rat

The next question was if the number of MCs in the cochlear tissue remains constant or if it changes during the postnatal development of cochlea. Comparative analyses demonstrated an age-dependent decrease of the cochlear MC numbers. The flat cochlear preparations (Figure 5A) obtained from the inner ear of p1 Wistar rats contained significantly more MCs (*p* < 0.01) when compared to the explant of p9 Wistar rats. In detail, one day after birth, there were on average 17.6 MCs per flat preparation of cochlea (SD +/− 12.3). P3 animals had on average 14.7 MCs (SD +/− 8.6), p5 animals 9.8 MCs (SD +/− 6.2), p7 animals 4.7 MCs (SD +/− 3.5), and p9 animals had 2.6 MCs (SD +/− 2.1) (Figure 5B).

### 3.6. The Number of MCs in Cochlear Explants Changes upon Exposure of Cochlear Explants to Cisplatin

In the last part of the experiments, the impact of cisplatin on the presence of MCs was investigated. Cisplatin is an ototoxic medication triggering hearing loss in a significant proportion of cancer patients via induction of hair cell loss in high frequencies. In laboratory settings, cochlear explants obtained from experimental animals (mice or rats) are commonly used to study in vitro the mechanisms of ototoxicity in the hair cells. However, cisplatin can affect other cells in the cochlea, such as marginal cells, perivascular resident macrophage-like melanocytes, and basal cells of the stria vascularis [35]. Therefore, we wanted to determine the effect of cisplatin exposure on the cochlear MCs. Cochlear explants obtained from p5 Wistar rats were exposed for 24 h to various concentrations of cisplatin (10, 20, and 40 µM). After exposure, the flat cochlear preparations were stained with avidin to visualize and score the MCs and phalloidin to visualize the hair cells. The concentration of 10 µM cisplatin did not cause significant hair cell loss (Table 2) nor induce substantial changes in the MC numbers (Figure 6). Cisplatin used at 20 µM induced a significant hair cell loss (about 50%) and was associated with a higher number of cochlear MCs. Cisplatin at the concentration of 40 µM induced a loss of about 70% of the hair cells. Under these conditions, significantly lower numbers of MCs were observed in the cochlea.

## 4. Discussion

The current study demonstrates that under normal physiological conditions, MCs are present in the cochleae of rats and mice. The MCs were detected in modiolus, spiral limbus, and close to the Reissner’s membrane from the side of the scala vestibuli. No MCs were identified in the organ of Corti. The MCs were located within the cochlear tissues, consistent with the status of the resident cells. The numbers of cochlear MCs scored in the flat preparations containing basilar membrane, the organ of Corti, spiral limbus, and the spiral ganglion neurons decreased during the postnatal development of Wistar rats. Furthermore, exposure to cisplatin significantly affected their numbers. Our data complement and extend earlier scarce reports on the existence of MCs in the inner ear [28,30].

The identity of cochlear MCs was verified using various methods. The first method was a double-staining with avidin and an antibody against CD117. Avidin is a protein with a high affinity to heparin [36]. Glycosaminoglycan abundantly presents in the granule of MCs [37], whereas CD117 is a receptor tyrosine kinase present on a surface of MCs, hematopoietic stem cells, and various types of cancer cells, confirming its role as a proto-oncogene [38]; it is also detected on melanocytes. Melanocytes that migrate during embryonal development into stria vascularis contribute to cochlear homeostasis by participation in potassium recycling, thus preserving the endocochlear potential [39]. Therefore, it could be possible that the CD117-positive staining identified cochlear melanocytes (intermediate cells in the stria vascularis). However, melanocytes do not contain heparin detected by avidin, and for the experiments when CD117 detection was employed, such as immunofluorescence in the flat cochlear preparations or the Western blot, only the stria vascularis-free preparations were used. Other molecules used to verify the identity of the cochlear MCs were the mast cells tryptase and chymase [10]. Both proteases participate in tissue remodeling, wound healing, and allergic reactions [40]. Although the roles of tryptase and chymase have not yet been studied in cochlear physiology or pathology, one can speculate that these mast cell proteases could participate in cochlear tissue remodeling during the growth and development of cochlea or after injury. Lastly, the expression of high-affinity immunoglobulin epsilon receptor subunit-alpha FcεRIα on the surface of the cochlear avidin-positive cells ultimately confirms the identity of these cells as MCs.

The biological purpose of cochlear MCs is currently unclear. It would be tempting to speculate that the MCs play a dual role in the auditory periphery. Because the primary physiological function of mast cells is to control tissue homeostasis [41], the first hypothetical task of cochlear MCs could be participation in homeostasis and functional maturation of rodent cochlea. The second role of cochlear mast cells could be participation in innate and adaptive immunity [42,43]. Past research, which was restricted to the endolymphatic sac, demonstrated that the sensitization of guinea pigs with allergens leads to the IgE-mediated degranulation of MCs to the perisaccular connective tissues, accompanied by endolymphatic hydrops [30]. In addition, few clinical studies describing the presence of MCs in perisaccular connective tissue of the inner ear explain the clinical association between allergy and inner ear disorders [28,29,30]. The degranulation of MCs can be triggered not only by IgE antibodies but also by other stimuli (e.g., temperature, neurotrophins) to induce a spill of cytokines [13], proteases [44], heparin, and histamine [45]. Histamine receptors have been identified in the inner ear of guinea pigs [46], rats [47], mice [48], and in the human inner ear [49], including the cochlear tissues. In guinea pigs, histamine applied to the cochlear perilymph influenced the compound action potential in a dose-dependent manner [46]. In contrast, in the vestibular organ of frogs, histamine has increased the evoked afferent firing rate of the ampullar nerve, consistent with its ability to induce vertigo [50].

The interesting aspect revealed by the present study is the significant decrease in the numbers of cochlear MCs observed in the flat cochlear preparations. This reduction occurred during the maturation of rodent cochlea (p1 to p9). This finding might be explained by recent studies demonstrating the involvement of MCs in the physiological development of the cornea of mice by participation in the tissue vascularization and maturation of corneal nerve fibers [51]. The numbers of corneal MCs observed during embryonic development gradually increased from E14.5 to E20.5, followed by a decrease and a complete absence from day 13 after birth. Based on the above knowledge, it is tempting to speculate that also in the cochlea, the MCs might contribute to the maturation of spiral ganglion fibers, and after the final maturation stage is achieved, the numbers of MCs decrease. Supporting this hypothesis is that MCs produce neurotrophins such as nerve growth factor (NGF) [52], which is essential for proper postnatal development of spiral ganglion neurons [53]. Moreover, MCs produce histamine and serotonin, and both neurotransmitters could contribute to the final tuning of the peripheral auditory system during the first two weeks of the rodent’s life. Such a notion is supported by an observation made in the CNS about the mast-cell-derived serotonin being essential for the maturation of hippocampal neurons [54]. Further, in the thymus, duodenum, and the mammary gland, the presence and numbers of MCs are associated with the proper function and development of these organs [55], which could also be the case for cochlea. Helpful for understanding the physiology of cochlear MCs would be the investigation of their presence, numbers, and function in the prenatal cochlea.

The last aspect of the presented study regarded the impact of cisplatin on the cochlear MCs. Cisplatin, a common cytostatic medication used to treat various types of tumors, induces toxicity in the kidney and the auditory periphery [56]. The ototoxic mechanism attributed to cisplatin involves elevated reactive oxygen species leading to hair cell loss, particularly in the basal region responsible for the transduction of high frequencies. Not only hair cells but also other cell types (e.g., in the stria vascularis) can be negatively affected by cisplatin. Here, in addition to the expected hair cell loss (Table 2), we observed dose-dependent fluctuation in the cochlear MC numbers upon 24 h exposure to 10, 20, or 40 µM cisplatin. Cisplatin at the concentration of 10 µM did not cause either hair cell loss or change of MC numbers. At the concentration of 20 µM, significant hair cell loss was observed, and a substantial increase in MC numbers was noted. Exposure to cisplatin at the concentration of 40 µM led to even greater hair cell loss and a significant decrease of MCs visualized with avidin. Cisplatin used at nanomolar concentrations was shown to induce histamine secretion from the murine MCs [57] but could also induce apoptosis in the human HMC-1 mast cell line [58] when used at the concentration of 10 µg/mL (33.3 µM), which is within the range used in the present work. Therefore, it can be assumed that the effect of cisplatin on the MCs ranges from degranulation to apoptosis, depending on its concentration. Notably, the degranulation of MCs is associated with the release of many neuronal and immune mediators into the immediate tissue environment. Such a release (for instance, of TNF-alpha) could amplify the cisplatin-induced damage. The degranulated (but not dead) MCs can no longer be detected by avidin, but after a certain time, the regranulation process would make them visible when using avidin staining. Further experiments should determine the effects of cisplatin on the cochlear MCs and clarify if MCs amplify cisplatin-induced ototoxicity.

Our work is not free of pitfalls. The first drawback is using the cochlear explants/flat preparations for MC scoring. That method left the numbers of MCs in the lateral wall and the modiolus unknown. However, using the cryosections (or paraffin sections) would require extreme workload and costs. Therefore, using new technologies such as iDisco [59] could, in the future, yield more information on the numbers and distribution of cochlear MCs in the intact tissues. Another drawback of this study was that we did not address the possible sexual dimorphism or inbred and outbred issues. However, prospective studies designed to answer the question about sex- or strain-related differences in the numbers or localization of cochlear MCs and their functional role in cochlear physiology should clarify this query.

## 5. Conclusions

In conclusion, we report here for the first time that MCs are present in the tissues of rodent cochleae. The localization of MCs in the cochlea included modiolus, spiral limbus, and spiral ligament. The numbers of MCs present in flat preparations (basilar membrane, the organ of Corti, spiral limbus, and spiral ganglion neurons) decreased during the postnatal development of Wistar rats. Furthermore, the relative levels of CD117 in flat preparations changed over time, being the greatest on p5. Lastly, cisplatin affected the numbers of cochlear MCs in the p5 cochlear flat preparations. Further experiments should shed more light on the function of MCs in cochlear development, homeostasis, and pathology.

## Figures and Tables

**Figure 1 brainsci-10-00697-f001:**
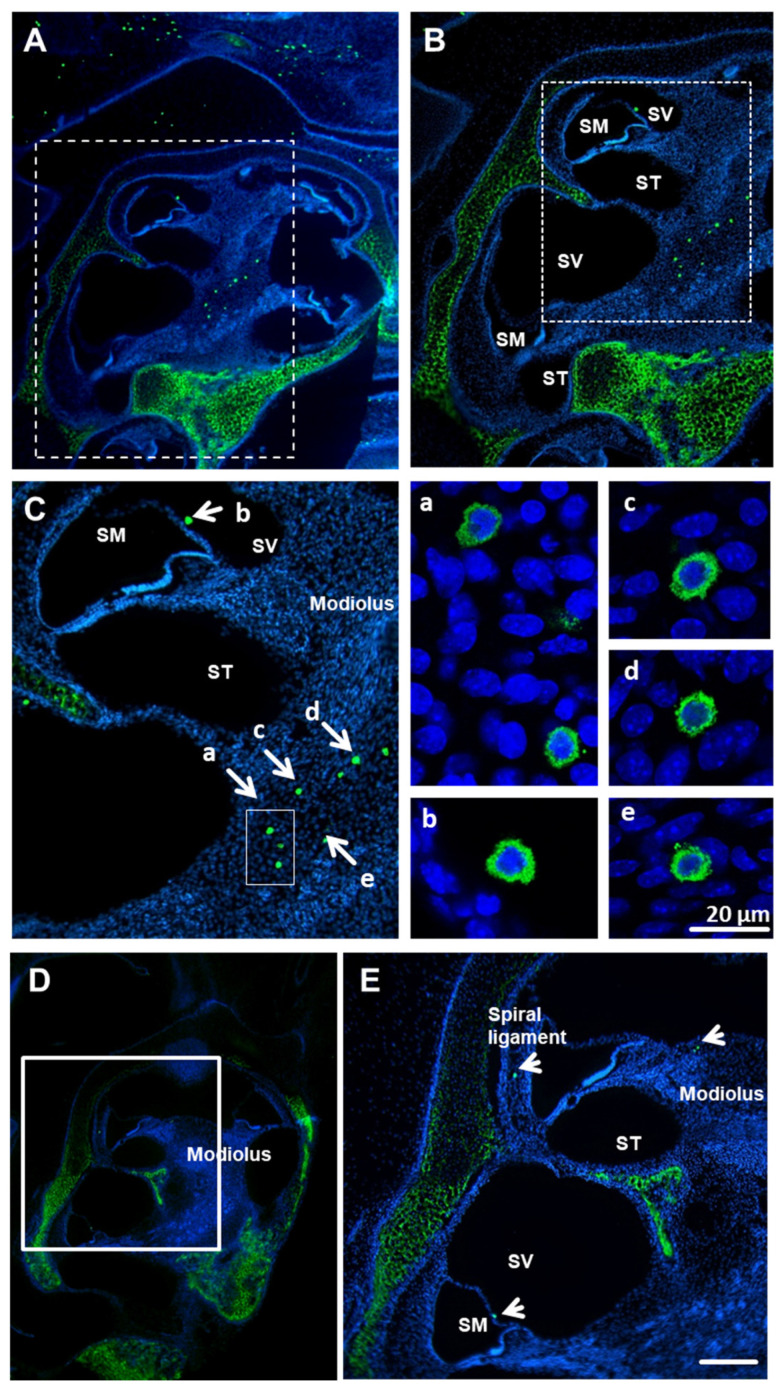
Avidin-positive MCs (mast cells) are present in the cochleae of mice and rats. MOUSE: Low (**A**), medium (**B**) and high (**C**) magnification of whole cochlear (p3 mice, *n* = 6) paraffin-embedded sections (10 µm) labeled with avidin-AlexaFluor-488 (green) and DAPI (blue). Avidin-positive MCs in scala vestibuli and modiolus; nuclei counterstained with DAPI. (**C**). Enlarged images of various cochlear MCs (**a**–**e** from panel **C**); scale bar 20 µm. RAT: Low (**D**) and medium (**E**) magnification of whole cochlear (p3 rat, *n* = 6) paraffin-embedded sections (10 µm) labeled with avidin-AlexaFluor-488 (green) and DAPI (blue). Avidin-positive MCs are visible in modiolus and the spiral ligament. Scale bar 50 µm. SM—scala media, ST—scala tympani, SV—scala vestibuli. There is visible staining of cartilage due to the known affinity of avidin to cartilage tissues [34]. Low magnification of bright field (**F**) and fluorescent field (**G**) representing paraffin-embedded sections (10 µm) labeled with avidin-AlexaFluor-488 (green) and DAPI (blue). The micrograph of the bright field demonstrates that the mast cells are localized to the cochlear bone tissue, which is not visible when using fluorescence.

**Figure 2 brainsci-10-00697-f002:**
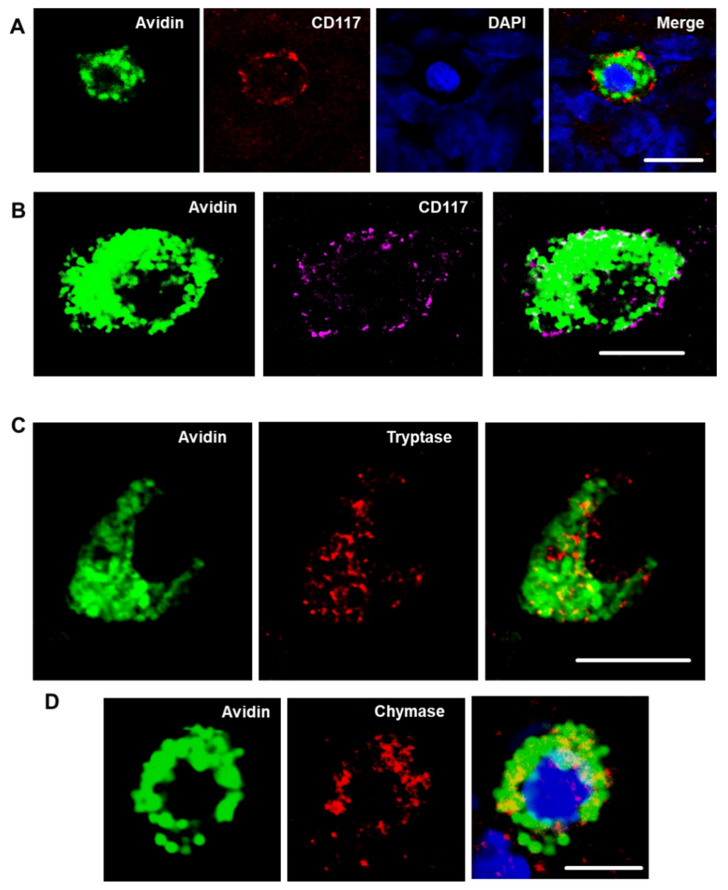
CD117, tryptase, and chymase are expressed by cochlear MCs. Representative confocal images of cochlear MCs that were immunopositive for CD117 (red / magenta) (**A**,**B**), tryptase (red) (**C**) and chymase (red) (**D**). MCs were detected with Alexa Fluor^®^ 488-conjugated avidin (green). Nuclei in (**A**,**D**) were counterstained with DAPI (blue). Images depicted from the modiolus region of medial turn from p3 rat cochlea. Images (**A**–**C**) were obtained from p3 Wistar rats; image (**D**) was obtained from p30 mouse. Scale bars 10 µm.

**Figure 3 brainsci-10-00697-f003:**
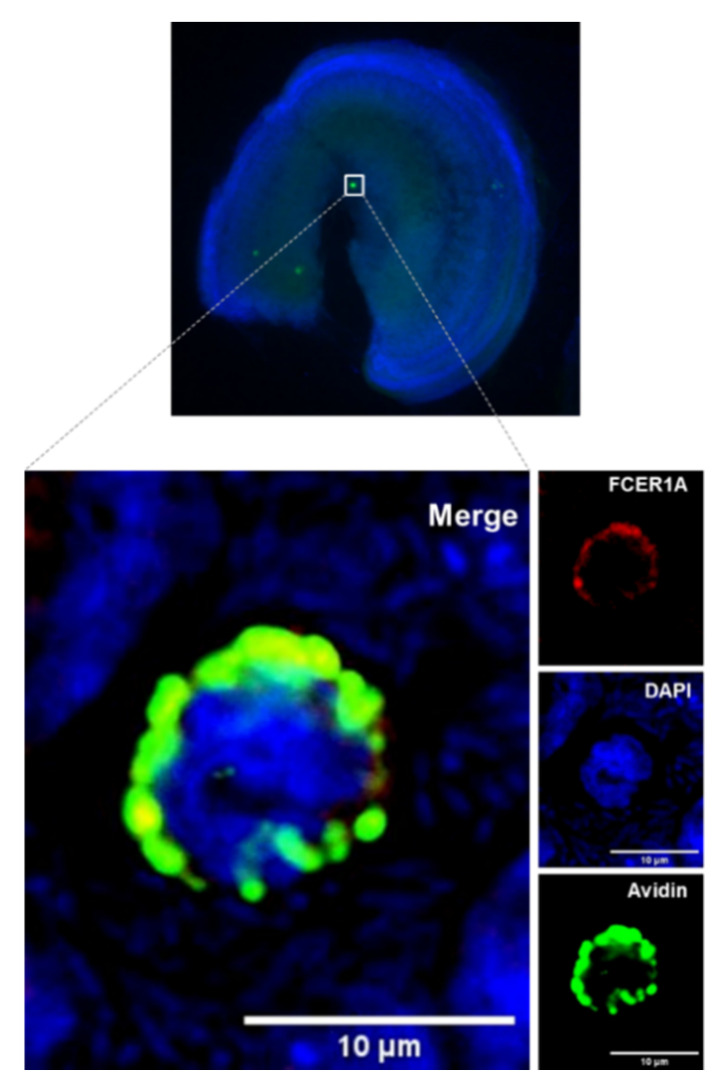
Cochlear MCs express the high-affinity immunoglobulin epsilon receptor subunit-alpha (FcεRIα). The image shows a cochlear explant of p3 mouse stained with an antibody against FcεRIα (red), avidin (green), and DAPI (blue).

**Figure 4 brainsci-10-00697-f004:**
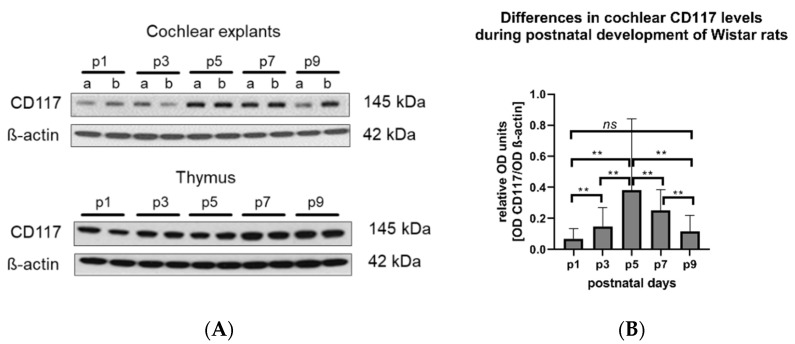
The levels of CD117 in the cochlea of Wistar rats vary during postnatal development. (**A**) Representative Western blot (WB) images showing CD117 (MW 142 kDa) expression in cochlear explants (**A** upper panel) and the thymus (**A** lower panel) isolated from naïve Wistar rats (p1–p9). β-actin (MW 42 kDa) was used as loading control; 6 µg proteins per lane were loaded. In the cochlear panel: a—apical, b—basal turn of the cochlea. In the thymus panel, each line corresponds to a different tissue sample. (**B**) Optical density (OD) of each sample (*n* = 16 cochleae for each developmental time point) was measured and expressed as the OD of CD117 divided by the OD of ß-actin from the same sample. The statistical analysis indicated significant differences between the CD117 expression on the consecutive developmental days (one-way ANOVA, F (4,75) = 4.819, ** *p* = 0.0016). *ns*: No statistical difference was detected between the OD of CD117 on p1 and p9 (Mann–Whitney test).

**Figure 5 brainsci-10-00697-f005:**
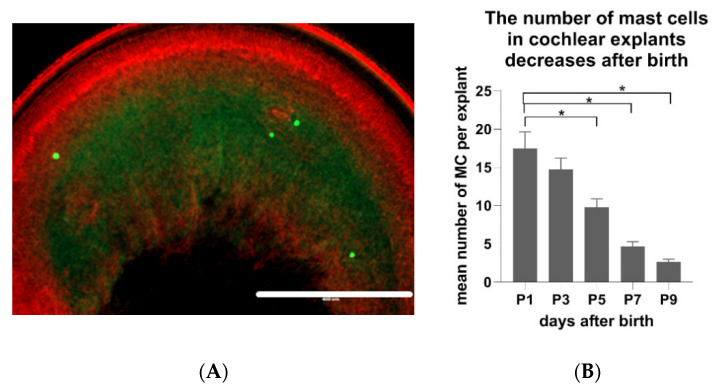
The number of cochlear MCs of Wistar rats decreases during postnatal development (p1–p9). (**A**) An exemplary micrograph of MCs visualized in the flat preparation of cochlear explants with avidin-AlexaFluor^®^ 488 and phalloidin-Texas red^TM^ (to stain filamentous actin). Scale bar 400 µm. (**B**) Quantification of MC numbers in the flat cochlear preparations of naïve p1–p9 Wistar rats. Presented are the average values, whiskers +/− SD. Significance: * *p* < 0.01 (one-way ANOVA). Sample size (cochleae): p1 *n* = 32, p3 *n* = 32, p5 *n* = 32, p7 *n* = 32, p9 *n* = 32.

**Figure 6 brainsci-10-00697-f006:**
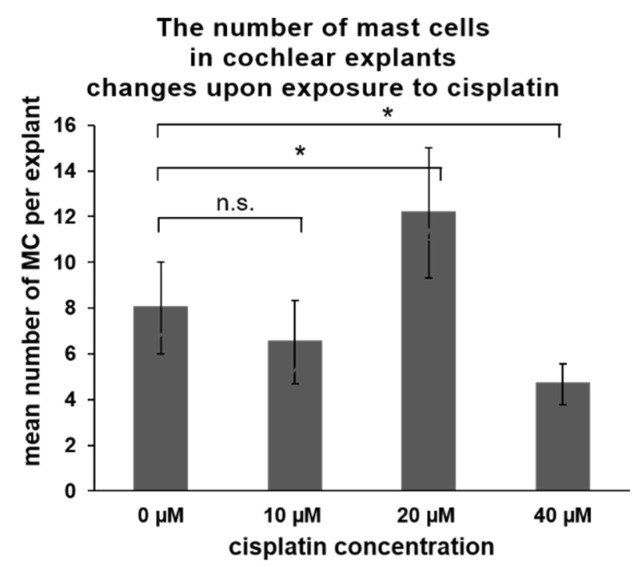
Cisplatin exposure-induced changes in a number of cochlear MCs of Wistar rats (p5). The explants (*n* = 6 per treatment) were exposed to various concentrations of cisplatin. The numbers of cochlear MCs were scored and are presented as an average value (whiskers +/− SD). A significant increase in the MC number was observed when 20 µM cisplatin was used, whereas a decrease was observed when 40 µM cisplatin was used. Significance: * *p* < 0.05, Student’s-t test; n.s.—not significant.

**Table 1 brainsci-10-00697-t001:** Primary and secondary antibodies and dyes.

	Target Molecule	Description/Isotype/Conjugate	Company	Catalog #	Working Dilution
Primary antibodies	c-Kit/CD117	Rabbit polyclonal/IgG	Thermo Scientific	PA5-16770	1:100
	MC chymase	Rabbit polyclonal/IgG	Cusabio	CSB-PA005599GA01HU	1:1000
	MC tryptase	Mouse monoclonal/ IgG_1_ (kappa light chain)	Santa Cruz	sc-59587	1:200
	High-affinity immunoglobulin epsilon Receptor subunit alpha (FcεRIα)	Rat polyclonal/IgG (whole molecule)	Cusabio	CSB-PA008532LA01HU-50	1:50
Secondary antibodies	Goat anti-Rat	Texas Red™-x (Ex = 595, Em = 615)	Thermo Scientific	T-6392	1:400
	Goat anti-Rabbit	Alexa Fluor^®^ 594 (Ex = 590, Em = 617)	Thermo Scientific	R 37117	1:300
	Goat anti-Rabbit IgG	Alexa Fluor^®^ 488 (Ex = 459, Em = 519)	Thermo Scientific	A 11001	1:400
	Goat anti-Mouse IgG	Alexa Fluor^®^ 594 (Ex = 590, Em = 617)	Thermo Scientific	A11005	1:400
	Goat anti-Mouse	Alexa Fluor^®^ 633 (Ex = 632, Em = 647)	Thermo Scientific	A 21050	1:1000
Fluorochromes & other reagents	Avidin	Alexa Fluor^®^ 488 (Ex = 459, Em = 519)	Invitrogen	A 21370	1:400
	DAPI (4‘,6 Diamidino-2-Phenylindole-Dihydrochloride)	(Ex = 364, Em = 454)	Sigma Aldrich	D 9542-5M6	1:10000
	Phalloidin	iFluor 594 (Ex = 590, Em = 617)	CytoPainter	ab176757	1:1500
	ProLong^®^ Gold (Antifade reagent)	Mounting solution	Invitrogen	P 36930	

**Table 2 brainsci-10-00697-t002:** Influence of cisplatin on the hair cell loss. Shown are the average numbers of intact inner and outer hair cells per 100 µm in the apical, medial, and basal parts of the explants. Significant differences compared to the control sample were calculated with one-way ANOVA; significance ** *p* < 0.01; n.s.—not significant.

	Inner Hair Cells			Outer Hair Cells		
Conditions	Apical	Medial	Basal	Apical	Medial	Basal
Control (*n* = 6)	9.56 ± 0.58	0.28 ± 1.02	10.00 ± 0.69	12.33 ± 0.54	11.72 ± 0.54	11.37 ± 1.33
10 µM cisplatin (*n* = 6)	8.78 ± 1.10 (n.s.)	9.11 ± 1.04 (n.s.)	9.83 ± 0.81 (n.s.)	12.20 ± 0.93 (n.s.)	11.56 ± 0.58 (n.s.)	11.02 ± 0.46 (n.s.)
20 µM cisplatin (*n* = 6)	6.17 ± 0.35 **	7.61 ± 0.92 **	4.94 ± 1.71 **	8.54 ± 1.32 **	9.31 ± 1.32 **	5.74 ± 0.31 **
40 µM cisplatin (*n* = 6)	4.78 ± 0.40 **	4.89 ± 2.74 **	3.83 ± 0.86 **	3.69 ± 1.84 **	4.87 ± 2.99 **	3.35 ± 1.02 **

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
