# Peer review of "Mast Cells in the Auditory Periphery of Rodents"

_brainsci, 2020, doi:10.3390/brainsci10100697_

Round 1
Reviewer 1 Report
This is a well performed study of mast cell activity in the mammalian cochlea. These cells are interesting since they could be involved in severe reactions damaging inner ear function. These cells can be observed in the perisaccular tissue around the guinea pig endolymphatic sac where they were decribed before using electron microscopy. Hence they were also seen activated in experimental hydrops. That these cells are present in the cochlea is interesting and raises several issue about their possible roles in inflammatory reactions.
The paper is well written and several techniques have been used. IHC is well performed and the images are of high quality. Western blot technique I cannot evaluate.
sentence 301 should be corrected
Ethical issues ok.
Author Response
sentence 301 should be corrected.
The sentence in line 301 was corrected.
Reviewer 2 Report
Major
This manuscript is a very interesting work that addresses a little-studied aspect of cochlear pathophysiology. The authors need to better explain some aspects. Specifically:
- Figure 1 shows a large number of apparently positive points in some areas, panel A, as the base of the modiolus, which do not appear to be MCs. There are also some dots in spaces (A) that are not identified but appear acellular (top outside the box). The authors should discuss two aspects: what are these positive points and the specificity of the technique used.
- The authors show that there are MCs in the cochlea but in low numbers and decreasing with age. The discussion, which is the weakest part of the work, does not address the significance of this reduction. It would be interesting to know if these cells are abundant in perinatal stages. At least the authors could elaborate whether these cells could play a role in late postnatal development during which there is a great remodeling in the cochlea.
- Finally, it is not clear why the response to cisplatin is studied, this part of the work is little elaborated and the results are not striking. Cisplatin doesn't work with a dose response, which doesn't make a lot of sense. The rationale for studying injury is not elaborated and the data is not excessively discussed. In general, the discussion is more a repetition of the results than a proper analysis of their importance or potential significance in the response to damage. The authors should elaborate the discussion further.
Minor
Well written text but with small errors that should be carefully revised and edited (example 3.1.4. title Levels of… change not changes)
There is a lack of homogeneity in the presentation.
Units must be separated by a space from the figures always (example page 11).
All axes must be labeled with title and units (example Figure 6)
Author Response
Major
This manuscript is a very interesting work that addresses a little-studied aspect of cochlear pathophysiology. The authors need to better explain some aspects. Specifically:
- Figure 1 shows a large number of apparently positive points in some areas, panel A, as the base of the modiolus, which do not appear to be MCs. There are also some dots in spaces (A) that are not identified but appear acellular (top outside the box). The authors should discuss two aspects: what are these positive points and the specificity of the technique used.
We added panels F and G to the Figure 1. These two panels (bright field and fluorescence) demonstrate that the mast cells are localized to the cochlear bone tissue, which is not visible when using fluorescence.
- The authors show that there are MCs in the cochlea but in low numbers and decreasing with age. The discussion, which is the weakest part of the work, does not address the significance of this reduction. It would be interesting to know if these cells are abundant in perinatal stages. At least the authors could elaborate whether these cells could play a role in late postnatal development during which there is a great remodeling in the cochlea.
We mention in the discussion (end of the second paragraph) the potential contribution of MC proteases to the tissue remodeling: “Although the role of tryptase and chymase was not yet studied in cochlear physiology or pathology, one can speculate that these mast cell proteases could participate in the cochlear tissue remodeling during the growth and development of cochlea or after injury. “
We also added the following in the discussion:
The interesting aspect revealed by the present study is the significant decrease in the numbers of cochlear MCs observed in the flat cochlear preparations. This reduction occurred during the maturation of rodent cochlea (p1 to p9). That finding might be explained by recent studies demonstrating the involvement of MCs in the physiological development of the cornea of mice by participation in the tissue vascularization and maturation of corneal nerve fibers [51]. The numbers of corneal MCs observed during embryonic development gradually increased from E14.5 to E20.5, followed by a decrease and a complete absence from day 13 after birth. Based on the above knowledge, it is tempting to speculate that also in the cochlea, the MCs might contribute to the maturation of spiral ganglion fibers, and after the final maturation stage is achieved, the numbers of MCs decrease. Supporting this hypothesis is that MCs produce neurotrophins such as nerve growth factor (NGF) [52], which is essential for proper postnatal development of spiral ganglion neurons [53]. Moreover, MCs produce histamine and serotonin, and both neurotransmitters could contribute to the final tuning of the peripheral auditory system during the first two weeks of rodent's life. Such a notion is supported by an observation made in the CNS about the mast cell-derived serotonin being essential for the maturation of hippocampal neurons [54]. Further, in the thymus, duodenum, and the mammary gland, the presence and numbers of MCs are associated with the proper function and development of these organs [55], which could also be the case for cochlea. Helpful for understanding the physiology of cochlear MCs would be the investigation of their presence, numbers, and function in the prenatal cochlea.
- Finally, it is not clear why the response to cisplatin is studied, this part of the work is little elaborated and the results are not striking. Cisplatin doesn't work with a dose response, which doesn't make a lot of sense. The rationale for studying injury is not elaborated and the data is not excessively discussed. In general, the discussion is more a repetition of the results than a proper analysis of their importance or potential significance in the response to damage. The authors should elaborate the discussion further.
We now justify our experiments as follows “However, cisplatin can affect other cells in the cochlea, such as marginal cells, perivascular resident macrophage-like melanocytes, and basal cells of the stria vascularis”. Therefore, we wanted to determine the effect of cisplatin exposure on the cochlear MCs.
We also revised the discussion as follows:
“Not only hair cells but also other cell types (e.g., in the stria vascularis) can be negatively affected by cisplatin. Here, in addition to the expected hair cell loss (Table 2), we observed dose-dependent fluctuation in the cochlear MC numbers upon 24h exposure to 10, 20, or 40 µM cisplatin. Cisplatin at the concentration of 10 µM has not caused either hair cell loss or change of MC numbers. At the concentration of 20 µM, significant hair cell loss was observed, and a substantial increase in MC numbers was noted. Exposure to cisplatin at the concentration of 40 µM led to even greater hair cell loss, and a significant decrease of MCs visualized with avidin. Cisplatin used at nanomolar concentrations was shown to induce histamine secretion from the murine MCs [57] but could also induce apoptosis in the human HMC-1 mast cell line [58] when used at the concentration of 10 µg/ml (33.3 µM), which is within the range used in the present work. Therefore, it can be assumed that the effect of cisplatin on the MCs ranges from degranulation to apoptosis, depending on its concentration. Notably, the degranulation of MCs is associated with the release of many neuronal and immune mediators into the immediate tissue environment. Such a release (for instance, of TNF-alpha) could amplify the cisplatin-induced damage. The degranulated (but not dead) MCs can no longer be detected by avidin, but after a certain time, the regranulation process would make them visible when using avidin staining. Further experiments should determine the effects of cisplatin on the cochlear MCs and clarify if MCs amplify cisplatin-induced ototoxicity. “
Minor
Well written text but with small errors that should be carefully revised and edited (example 3.1.4. title Levels of… change not changes)
Thank you for pointing this out. The paper was screened and revised.
There is a lack of homogeneity in the presentation.
The data is now presented in a homogenous manner.
Units must be separated by a space from the figures always (example page 11).
This has been corrected.
All axes must be labeled with title and units (example Figure 6)
All axes are now labeled.